# TeFlow: Enabling Multi-frame Supervision for Feed-forward Scene Flow Estimation

## Abstract

Self-supervised feed-forward methods for scene flow estimation offer real-time efficiency, but their supervision from two-frame point correspondences is unreliable and often breaks down under occlusions. Multi-frame supervision has the potential to provide more stable guidance by incorporating motion cues from past frames, yet naive extensions of two-frame objectives are ineffective because point correspondences vary abruptly across frames, producing inconsistent signals. In the paper, we present TeFlow, enabling multi-frame supervision for feed-forward models by mining temporally consistent supervision. TeFlow introduces a temporal ensembling strategy that forms reliable supervisory signals by aggregating the most temporally consistent motion cues from a candidate pool built across multiple frames. Extensive evaluations demonstrate that TeFlow establishes a new state-of-the-art for self-supervised feed-forward methods, achieving performance gains of **up to 33%** on the challenging Argoverse 2 and nuScenes datasets. Our method performs on par with leading optimization-based methods, yet speeds up **150** times. The source code and model weights will be released upon publication.

## 1 Introduction

Scene flow determines the 3D motion of each point between consecutive point clouds as visualized in Figure 1a. By providing a detailed characterization of object motion, scene flow could benefit downstream tasks such as motion prediction Najibi et al. (2022), dynamic object reconstruction Chodosh et al. (2025); Zhang et al. (2025a), and occupancy flow prediction Yang et al. (2024). Accurate scene flow prediction enables autonomous agents to capture the underlying environmental dynamics during observation Li et al. (2025); Jia et al. (2024).

To overcome the high cost of manual annotation required by supervised methods Zhang et al. (2024a); Jund et al. (2021); Khoche et al. (2025); Luo et al. (2025), the field has increasingly shifted towards self-supervised learning, which exploits geometric and temporal consistency across frames without requiring ground-truth labels. Existing self-supervised approaches fall into two categories: (1) Optimization-based methods Vedder et al. (2024b); Hoffmann et al. (2025) achieve high accuracy by enforcing long-term multi-frame constraints but suffer from substantial optimization latency, making them unsuitable for real-time deployment. As shown in Figure 1c, the optimization of such methods can take hours and days for a single scene. (2) Feed-forward methods Zhang et al. (2024b); Lin et al. (2025) achieve high efficiency by generating results in a single forward pass, however, their accuracy is limited by unstable training objectives derived from only two consecutive frames. For example, as shown in Figure 1a, when depicting objects (e.g., pedestrians), occlusions often cause missing points between frames, preventing consistent motion guidance and leading to incorrect flows. In addition, two-frame supervision is also vulnerable to sensor noise, sparse observations, and ambiguity in curved or articulated motion. Leveraging information from multiple frames mitigates these issues and provides a more stable and temporally consistent supervisory signal.

However, introducing additional frames into feed-forward training is non-trivial. As shown in Figure 1b, the direction of the two-frame supervisory signal varies drastically over time. Even when the underlying motion is smooth, two-frame estimates fluctuate sharply due to occlusions, noise, and missing points. Training with such temporally inconsistent signals prevents the model from learning coherent motion patterns and results in inaccurate scene flow. This highlights the importance of exploiting temporally consistent cues across multiple frames to provide effective supervision for

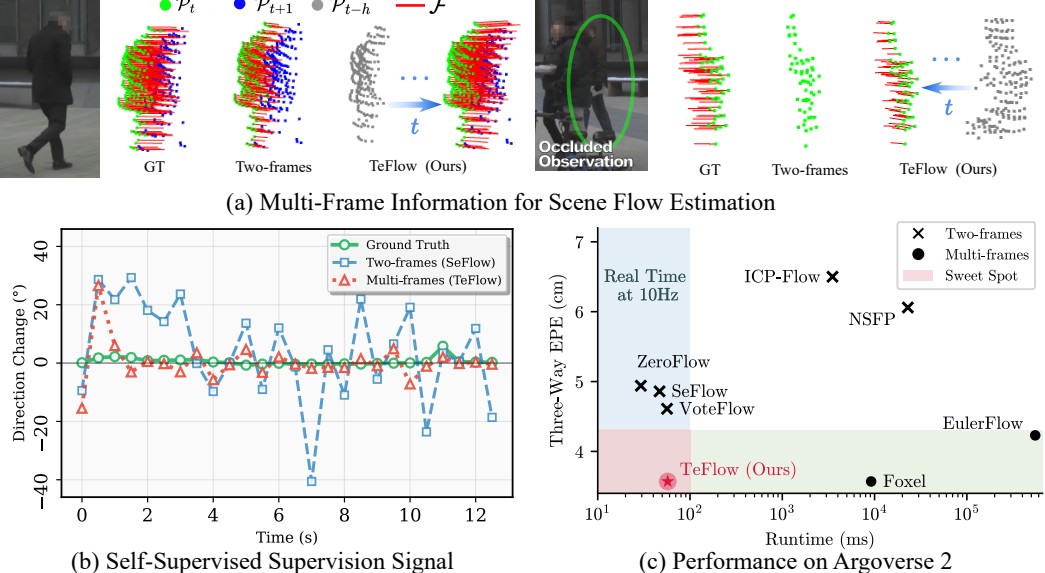

Figure 1: (a) Multi-frame supervision maintains stable guidance during occlusion by querying past frames, while two-frame supervision fails due to missing points. (b) Direction change of supervisory signals over time, reflecting their temporal consistency. The two-frame supervision Zhang et al. (2024b) exhibits abrupt variations with frequent direction shifts, while our five-frame TeFlow produces more stable signals that stay closer to the ground truth. (c) Accuracy vs. Runtime. Prior feed-forward methods are fast but less accurate, while optimization-based methods are accurate but too slow. TeFlow achieves both real-time speed and high accuracy.

feed-forward models. To achieve this, we propose TeFlow, a novel multi-frame feed-forward framework that mines consistent motion signals across time. TeFlow introduces a temporal ensembling strategy that constructs a pool of motion candidates across multiple frames and applies a voting scheme to aggregate the most consistent ones. The resulting consensus motions form a robust supervisory signal, enabling feed-forward models to achieve high-accuracy scene flow estimation while maintaining real-time efficiency.

Our contributions can be summarized as follows:

- We leverage temporally-consistent supervisory signals for self-supervised scene flow estimation by constructing a motion candidate pool from multiple frames and then optimizing the consensus motion via a voting scheme.
- By integrating our objective function, TeFlow becomes the first approach to unlock the potential of multi-frame network architectures in a real-time, self-supervised setting.
- We demonstrate through extensive experiments on the Argoverse 2 and nuScenes datasets that TeFlow achieves the state-of-the-art performance for real-time self-supervised methods, significantly narrowing the accuracy gap to slow optimization-based methods while maintaining real-time efficiency.

## 2 RELATED WORK

Scene flow estimation Vedula et al. (2005); Lang et al. (2023); Khatri et al. (2024); Jiang et al. (2024); Zhang et al. (2024d) has been a long-standing problem in computer vision. Our work builds upon advances in both supervised and, more importantly, self-supervised learning paradigms.

**Supervised Scene Flow** Early and many current state-of-the-art methods are trained in a fully supervised manner Wei et al. (2021); Wang et al. (2023); Liu et al. (2024); Zhang et al. (2024c). These approaches leverage large datasets with ground-truth flow annotations to train deep neural networks. Methods like FastFlow3D Jund et al. (2021), DeFlow Zhang et al. (2024a), and SSF Khoche et al. (2025) use voxel-based backbones to efficiently process large-scale point clouds, achieving high accuracy and real-time inference speeds. While powerful, these methods are fundamentally limited by their reliance on expensive, manually annotated data, which is difficult to scale and may not cover all real-world scenarios.

**Self-Supervised Scene Flow** To overcome the need for labeled data, self-supervised methods have gained significant interest. These methods can be broadly divided into two main strategies: optimization-based approaches and feed-forward approaches.

Optimization-based approaches fit a scene-specific model at test time. The pioneering NSFP Li et al. (2021) optimizes a small coordinate-based MLP for each two-frame pair. Follow-up works Li et al. (2023); Hoffmann et al. (2025) improve efficiency by replacing the MLP with representations like voxel grids or distance transforms. To achieve higher accuracy, the state-of-the-art method, EulerFlow Vedder et al. (2024b), reframes scene flow as the task of estimating a continuous ordinary differential equation over an entire sequence. By optimizing a neural prior against reconstruction objectives across many frames, it produces exceptionally accurate flow fields. However, this accuracy comes at a prohibitive computational cost, requiring from hours to days of optimization for a single sequence, making it unsuitable for any real-time application.

In contrast, feed-forward methods aim to train a single, generalizable network on a large unlabeled dataset, enabling real-time inference on new scenes. A prominent approach is knowledge distillation, exemplified by ZeroFlow Vedder et al. (2024a). This technique uses a slow but accurate optimization-based 'teacher' to generate pseudo-labels for a fast 'student' network. However, this label generation process requires 7.2 GPU months of computation, which limits its scalability and practical adoption. Other methods, such as SeFlow Zhang et al. (2024b), instead design the two-frame loss functions directly. SeFlow first classifies points as static or dynamic Duberg et al. (2024) and then applies tailored consistency losses to each group to improve learning. Despite their different strategies, these methods are still fundamentally trained using supervisory signals derived from only two consecutive frames.

**Multi-frame Architectures** Independent of the training paradigm, network architectures have evolved to better capture temporal information. Models like Flow4D Kim et al. (2025) introduce an explicit temporal dimension and use 4D convolutions Choy et al. (2019) to process sequences of voxelized point clouds. Taking a different approach to efficiency, DeltaFlow Zhang et al. (2025b) introduces a computationally lightweight '$\Delta$ scheme' that directly computes the difference between voxelized frames. This avoids the feature expansion common in other multi-frame methods and maintains a constant input size regardless of the number of frames. This architectural trend shows a clear recognition in the community that temporal context is crucial for accurate motion estimation. However, when trained with self-supervision, these powerful backbones are still bottlenecked by the current two-frame-based supervision objectives, preventing them from reaching their full potential.

## 3 PRELIMINARIES

**Problem Formulation** Given a continuous stream of LiDAR point clouds, our goal is to train a feed-forward network $\Phi_\theta$ that estimates the scene flow vector field Zhang et al. (2025b). For a given frame $\mathcal{P}_t \in \mathbb{R}^{N_t \times 3}$, the network predicts its flow $\mathcal{F} \in \mathbb{R}^{N_t \times 3}$ toward the subsequent frame $\mathcal{P}_{t+1} \in \mathbb{R}^{N_{t+1} \times 3}$. The scene flow $\mathcal{F}$ is decomposed into two parts: ego-motion flow $\mathcal{F}_{\text{ego}}$ induced by the movement of the vehicle, and residual flow $\mathcal{F}_{\text{res}}$, caused by dynamic objects in the environment. Since ego-motion can be obtained directly from odometry, the network is trained only to estimate the residual flow. Formally, the network learns the mapping:

$$\Phi_\theta : \left\{ \mathbf{T}_{\text{ego}}^{t-h \to t+1} \mathcal{P}_{t-h}, \ldots, \mathbf{T}_{\text{ego}}^{t \to t+1} \mathcal{P}_t, \mathcal{P}_{t+1} \right\} \to \mathcal{F}_{\text{res}}, \tag{1}$$

where $\mathbf{T}_{\text{ego}}^{t' \to t+1} \in \mathbb{R}^{4 \times 4}$ is the odometry transformation matrix from time $t'$ to $t+1$, aligning all past point clouds to the coordinate frame of $\mathcal{P}_{t+1}$.

**Self-Supervised Training Paradigm** To train $\Phi_\theta$ without labeled data, we adopt a self-supervised paradigm that derives supervisory signals directly from the input sequence. Following Zhang et al. (2024b), point clouds are first segmented into static and dynamic regions ($\mathcal{P}_{\cdot,s}, \mathcal{P}_{\cdot,d}$). Static points $\mathcal{P}_{\cdot,s}$ are supervised with a near-zero flow loss. Dynamic points $\mathcal{P}_{\cdot,d}$ are further partitioned into clusters $\mathcal{C} = \{\mathcal{C}_1, \mathcal{C}_2, \ldots, \mathcal{C}_{N_c}\}$, where $N_c = |\mathcal{C}|$ is the number of dynamic clusters. Each cluster is assumed to undergo a shared rigid motion and is trained with a rigidity loss, i.e., dynamic cluster loss, that enforces coherent motion within the group. Prior work Zhang et al. (2024b) derives this loss from two-frame correspondences, which are often noisy and provide unstable supervision.

Figure 2: **An overview of the TeFlow**, a multi-frame feedforward scene flow estimation pipeline, shown in the top row. Our self-supervised pipeline tackles the main challenge of deriving reliable supervision $\bar{\mathbf{f}}$ from dense multi-frame inputs. For each dynamic cluster $\mathcal{C}_j$, we create a motion candidate pool $\mathcal{F}_{\mathcal{C}_j}$ (red arrows) from multi-frame geometry and network predictions $\hat{\mathbf{f}}_{\mathcal{C}_j}$. This pool is then aggregated as final cluster-level supervision $\bar{\mathbf{f}}_{\mathcal{C}_j}$ through weighted reliability voting, where $\mathbf{M}$ indicates inter-candidate consistency and $\mathbf{w}$ represents magnitude-based reliability.

## 4 METHOD: TEFLOW

To move beyond the limits of two-frame supervision and achieve both high accuracy and efficiency, we propose TeFlow, a multi-frame feed-forward framework illustrated in Figure 2, which generates stable supervisory signals through temporal ensembling of consistent motions across frames.

### 4.1 TEMPORAL ENSEMBLING FOR DYNAMIC CLUSTERS

TeFlow aims to assign each dynamic cluster $\mathcal{C}_j$ a reliable supervision target $\bar{\mathbf{f}}_{\mathcal{C}_j} \in \mathbb{R}^{1 \times 3}$ that reflects its true motion. A naive extension from two-frame to multi-frame supervision is unreliable, since frame-to-frame correspondences often vary abruptly and introduce conflicting signals (as shown in Figure 1b). To address this, TeFlow introduces a temporal ensembling approach that first constructs a pool of motion candidates across the temporal window, capturing multiple hypotheses, and then forms a robust supervision signal by selecting and weighting only the most consistent motions. The approach consists of two stages: (i) generate a diverse pool of motion candidates across the temporal window, and (ii) aggregate the target motion via a weighted voting scheme.

**Motion Candidate Generation.** This stage aims to build a candidate pool from which a reliable supervisory target can be aggregated for each cluster $\mathcal{C}_j$. Each candidate is represented by a single 3D motion vector. The pool combines two complementary sources: internal and external candidates, which together balance stability with data-driven evidence.

The internal candidate $\hat{\mathbf{f}}_{\mathcal{C}_j}$ serves as an anchor that stabilizes the supervisory signal and keeps training grounded in the evolving state of the model. It is obtained from the current estimate from the network $\Phi_\theta$, computed as the average flow over all points in the cluster,

$$\hat{\mathbf{f}}_{\mathcal{C}_j} = \frac{1}{|\mathcal{C}_j|} \sum_{\mathbf{p}_i \in \mathcal{C}_j} \hat{\mathbf{f}}_i, \tag{2}$$

where $|\mathcal{C}_j|$ is the number of point in the cluster and $\hat{\mathbf{f}}_i \in \mathcal{F}_{\text{res}}$ is the network estimation for point $\mathbf{p}_i$.

The external candidates $\mathbf{f}_{\mathcal{C}_j,k}^{t'}$ represent geometry-based motion hypotheses for the cluster. They aim to approximate how the cluster might actually move by exploiting information from neighboring frames. To construct them, we compare the cluster $\mathcal{C}_j$ at time $t$ with the dynamic points $\mathcal{P}_{t',d}$ from each of the other frames $t' \in \{t-h, \ldots, t-1, t+1\}$. For every pair of frames $(t, t')$, we establish correspondences by finding, for each point $\mathbf{p}_i \in \mathcal{C}_j$, its nearest neighbor in $\mathcal{P}_{t',d}$. Among these correspondences, we retain the Top-$K$ with the largest displacement magnitudes, as they are more likely to capture meaningful motion rather than noise.

Since different frames $t'$ are separated from $t$ by varying time intervals, the displacements are normalized by the temporal gap $(t'-t)$. The normalized motion vector for frame $t'$ and the $k$-th selected correspondence is defined as:

$$\mathbf{f}_{\mathcal{C}_j,k}^{t'} = \frac{\mathcal{NN}(\mathbf{p}_k, \mathcal{P}_{t',d}) - \mathbf{p}_k}{t' - t}, \qquad (3)$$

where $\mathcal{NN}(\cdot)$ denotes nearest-neighbor search and $\mathbf{p}_k$ is the $k$-th Top-$K$ source point.

Finally, we combine the internal candidate with all external candidates from the temporal window to form the complete candidate pool:

$$\mathcal{F}_{\mathcal{C}_j} = \{\hat{\mathbf{f}}_{\mathcal{C}_j}\} \cup \{\mathbf{f}_{\mathcal{C}_j,k}^{t'} \mid t' \in \{t-h, \ldots, t-1, t+1\},\ k \in \{1, \ldots, K\}\} \qquad (4)$$

This pool contains a total of $1 + K(h+1)$ candidates, each candidate $\mathbf{f}_i \in \mathbb{R}^{1\times 3}$. By uniting stability from the internal estimate with motion evidence from external correspondences, the pool provides a strong foundation for consensus in the subsequent voting stage.

**Candidate Voting and Flow Aggregation.** With the candidate pool constructed, the next step is to derive a stable cluster-level flow. Since the pool still contains a mix of useful and noisy motion vectors, selecting one directly could lead to unstable supervision. To obtain a reliable estimate, we aggregate candidates based on two criteria: (i) their agreement with others in the pool, and (ii) their own reliability.

The first criterion, agreement, captures which flows reinforce each other, ensuring that the final decision reflects collective support. It is measured through a consensus matrix $\mathbf{M} \in \mathbb{R}^{(1+K(h+1))\times(1+K(h+1))}$. Each entry $\mathbf{M}_{ab}$ indicates whether two candidates $\mathbf{f}_a$ and $\mathbf{f}_b$ are directionally consistent, determined by their cosine similarity $\tau_{\cos}$:

$$\mathbf{M}_{ab} = \begin{cases} 1 & \text{if } \dfrac{\mathbf{f}_a \cdot \mathbf{f}_b}{\|\mathbf{f}_a\|\|\mathbf{f}_b\|} > \tau_{\cos}, \\ 0 & \text{otherwise.} \end{cases} \qquad (5)$$

The second criterion, reliability, reflects how trustworthy each candidate is and therefore how much influence it should have on the final flow. It is encoded in a weight vector $\mathbf{w} = [w_1, \ldots, w_{1+K(h+1)}]^T$, where the weight of candidate $\mathbf{f}_i$ is defined as

$$w_i = \gamma^{m_i}\left(1 + \|\mathbf{f}_i\|_2^2\right). \qquad (6)$$

Here, $\gamma \in (0,1]$ is a temporal decay factor that prioritizes candidates from more recent frames, and $m_i$ is the time offset of $\mathbf{f}_i$, with $m_i = 0$ for the internal candidate and $m_i = |t'-t|$ for external ones. The magnitude term $\|\mathbf{f}_i\|_2^2$ further emphasizes larger displacements, which provide clearer motion cues than near-zero flows. This design encourages candidates with clearer motion cues to obtain higher weights and greater influence in the voting and aggregation stage.

With both agreement and reliability defined, we combine them to identify the most representative flow in the pool, referred to as the consensus winner. It is obtained as

$$a^\dagger = \underset{i\in\{1,\ldots,1+K(h+1)\}}{\arg\max} \mathbf{S}_i, \quad \text{where } \mathbf{S} = \mathbf{M}\mathbf{w}. \qquad (7)$$

Here, each element $\mathbf{S}_i$ aggregates the reliability weights of all candidates that agree with the $i$-th one, so a higher score means that a candidate is supported by more reliable neighbors. The index $a^\dagger$ therefore corresponds to the candidate with the strongest overall support.

Rather than relying only on this single winner, we further stabilize the supervision by taking a weighted average of flow candidates that are directionally consistent with the consensus winner:

$$\bar{\mathbf{f}}_{\mathcal{C}_j} = \frac{\sum_b \mathbf{M}_{a^\dagger b} w_b \mathbf{f}_b}{\sum_b \mathbf{M}_{a^\dagger b} w_b}. \qquad (8)$$

This averaging step preserves the reliability of the winner while incorporating supportive evidence from consistent candidates, mitigating the effect of noise and producing a stable supervisory target from both model predictions and multi-frame geometric evidence. As illustrated in Figure 1b, this strategy yields supervisory signals that are significantly more consistent than those from two-frame supervision. These signals $\bar{\mathbf{f}}_{\mathcal{C}_j}$ are then used to define our training objectives.

### 4.2 TRAINING OBJECTIVE

Building on previous two-frame approaches Zhang et al. (2024b), we define a dynamic cluster loss $\mathcal{L}_{\text{dcls}}$ using the supervision $\bar{\mathbf{f}}_{\mathcal{C}_j}$. The basic form is a *point-level* L2 loss, computed between the model predictions and the supervisory targets and averaged over all points in all dynamic clusters. However, as large objects contain more points, their losses dominate the training process, which biases the optimization and suppresses small objects. To solve the problem, we introduce a *cluster-level* loss term. Specifically, this term first averages the L2 error within each cluster and then averages across clusters, ensuring that small objects contribute fairly rather than being overshadowed by larger ones. The full dynamic cluster loss is the sum of the point-level and cluster-level terms:

$$\mathcal{L}_{\text{dcls}} = \underbrace{\frac{1}{|\mathcal{P}_{\mathcal{C}}|} \sum_j \sum_{\mathbf{p}_i \in \mathcal{C}_j} \|\hat{\mathbf{f}}_i - \bar{\mathbf{f}}_{\mathcal{C}_j}\|_2^2}_{\text{Point-level Term}} + \underbrace{\frac{1}{N_c} \sum_j \left( \frac{1}{|\mathcal{C}_j|} \sum_{\mathbf{p}_i \in \mathcal{C}_j} \|\hat{\mathbf{f}}_i - \bar{\mathbf{f}}_{\mathcal{C}_j}\|_2^2 \right)}_{\text{Cluster-level Term}}, \tag{9}$$

where $|\mathcal{P}_{\mathcal{C}}|$ is the total number of points across all dynamic clusters and $N_c$ is the number of clusters.

In addition to our proposed $\mathcal{L}_{\text{dcls}}$, we adopt two auxiliary losses from prior work Zhang et al. (2024b); Vedder et al. (2024b). The *static loss* $\mathcal{L}_{\text{static}}$ Zhang et al. (2024b) penalizes non-zero residual flow on background points $\mathcal{P}_{t,s}$, since their motion is already explained by ego-motion of the vehicle. The *geometric consistency loss* $\mathcal{L}_{\text{geom}}$ applies multi-frame Chamfer and dynamic Chamfer distances to ensure that the source point cloud, warped by the predicted flows, aligns with neighboring frames.

Together, these losses ensure that the network learns from reliable cluster-level supervision, respects static background constraints, and preserves global geometric consistency across time. The overall training objective is the sum of all three losses:

$$\mathcal{L}_{\text{total}} = \mathcal{L}_{\text{dcls}} + \mathcal{L}_{\text{static}} + \mathcal{L}_{\text{geom}}. \tag{10}$$

### 4.3 IMPLEMENTATION DETAILS

We build TeFlow on top of the multi-frame DeltaFlow backbone Zhang et al. (2025b). Static and dynamic segmentation for training is provided by DUFOMap Duberg et al. (2024), and dynamic clusters are pre-computed using HDBSCAN Campello et al. (2013). The main hyperparameters of our method are as follows: a cosine similarity threshold of $\tau_{cos} = 0.7071$ (corresponding to a $45°$ angular difference), a Top-$K$ selection of $K = 5$ for external candidates, and a temporal decay factor of $\gamma = 0.9$. For the DeltaFlow backbone, we adopt its standard configuration, processing a $76.8 \times 76.8 \times 6$ m region represented as a $512 \times 512 \times 40$ voxel grid with $0.15$ m resolution. Training is performed for 15 epochs using the Adam optimizer with a learning rate of $0.002$ and a total batch size of 20, distributed across ten NVIDIA RTX 3080 GPUs. Each dataset requires approximately 15 to 20 hours of training. The source code and model weights will be released upon publication.

## 5 EXPERIMENTS

**Datasets** Experiments are conducted on two large-scale autonomous driving datasets: Argoverse 2 Wilson et al. (2021), collected with two roof-mounted 32-channel LiDARs, and nuScenes Caesar et al. (2020), which uses a single 32-channel LiDAR. Details on datasets description, preprocessing, and ground-truth flow estimation are provided in Section A.

**Evaluation Metrics** We follow the official Argoverse 2 benchmark and report three-way End Point Error (EPE) Chodosh et al. (2024) and Dynamic Bucket-Normalized EPE Khatri et al. (2024). *Three-way EPE* computes the unweighted average EPE over three categories: foreground dynamic (FD), foreground static (FS), and background static (BS). *Dynamic Bucket-Normalized EPE* normalizes the EPE by the mean speed within predefined motion buckets, providing a fairer comparison across different object classes. It evaluates four categories: regular cars (CAR), other vehicles (OTHER), pedestrians (PED.), and wheeled vulnerable road users (VRU). All evaluations are conducted within a $70 \times 70$ m area around the ego vehicle.

**Baselines** We compare TeFlow against both optimization-based and feed-forward self-supervised methods: NSFP Li et al. (2021), FastNSF Li et al. (2023), ZeroFlow Vedder et al. (2024a), ICPFlow Lin & Caesar (2024), SeFlow Zhang et al. (2024b), SeFlow++ Zhang et al. (2025a), EulerFlow Vedder et al. (2024b), VoteFlow Lin et al. (2025) and Floxels Hoffmann et al. (2025). To

Table 1: Performance comparisons on the Argoverse 2 test set leaderboard Argoverse2 (2025). TeFlow achieves state-of-the-art performance in real-time scene flow estimation. '#F' denotes the number of input frames. Runtime is reported per sequence (around 157 frames), with '-' indicating unreported values. Units are given in seconds (s') and minutes (m').

| Methods | #F | Runtime per seq | Three-way EPE (cm) ↓ | | | | Dynamic Bucket-Normalized ↓ | | | | |
| | | | Mean | FD | FS | BS | Mean | CAR | OTHER | PED. | VRU |
| --- | --- | --- | --- | --- | --- | --- | --- | --- | --- | --- | --- |
| Ego Motion Flow | - | - | 18.13 | 53.35 | 1.03 | 0.00 | 1.000 | 1.000 | 1.000 | 1.000 | 1.000 |
| *Optimization-based* | | | | | | | | | | | |
| FastNSF | 2 | 12m | 11.18 | 16.34 | 8.14 | 9.07 | 0.383 | 0.296 | 0.413 | 0.500 | 0.322 |
| NSFP | 2 | 60m | 6.06 | 11.58 | 3.16 | 3.44 | 0.422 | 0.251 | 0.331 | 0.722 | 0.383 |
| ICP-Flow | 2 | - | 6.50 | 13.69 | 3.32 | 2.50 | 0.331 | 0.195 | 0.331 | 0.435 | 0.363 |
| Floxels | 13 | 24m | **3.57** | 7.73 | **1.44** | **1.54** | 0.154 | 0.112 | 0.213 | **0.195** | 0.096 |
| EulerFlow | all | 1440m | 4.23 | **4.98** | 2.45 | 5.25 | **0.130** | **0.093** | **0.141** | **0.195** | **0.093** |
| *Feed-forward* | | | | | | | | | | | |
| ZeroFlow | 3 | 5.4s | 4.94 | 11.77 | 1.74 | 1.31 | 0.439 | 0.238 | 0.258 | 0.808 | 0.452 |
| SemanticFlow | 2 | - | 4.69 | 12.26 | **1.41** | **0.40** | 0.331 | 0.210 | 0.310 | 0.524 | 0.279 |
| SeFlow | 2 | 7.2s | 4.86 | 12.14 | 1.84 | 0.60 | 0.309 | 0.214 | 0.291 | 0.464 | 0.265 |
| VoteFlow | 2 | 13s | 4.61 | 11.44 | 1.78 | 0.60 | 0.289 | 0.202 | 0.288 | 0.417 | 0.249 |
| SeFlow++ | 3 | 10s | 4.40 | 10.99 | 1.44 | 0.79 | 0.264 | 0.209 | 0.272 | 0.367 | 0.210 |
| TeFlow (Ours) | 5 | 8s | **3.57** | 8.53 | 1.49 | 0.70 | **0.205** | **0.163** | **0.227** | **0.253** | **0.177** |

Table 2: Performance comparisons on the nuScenes validation set with a 10Hz LiDAR frequency. TeFlow achieves state-of-the-art accuracy in scene flow estimation. Runtime is reported per sequence (≈200 frames) using the same device.

| Methods | #F | Runtime per seq | Dynamic Bucket-Normalized ↓ | | | | | Three-way EPE (cm) ↓ | | | |
| | | | Mean | CAR | OTHER | PED. | VRU | Mean | FD | FS | BS |
| --- | --- | --- | --- | --- | --- | --- | --- | --- | --- | --- | --- |
| Ego Motion Flow | - | - | 1.000 | 1.000 | 1.000 | 1.000 | 1.000 | 12.34 | 35.94 | 1.07 | 0.00 |
| *Optimization-based* | | | | | | | | | | | |
| NSFP | 2 | 3.5m | 0.602 | 0.463 | 0.456 | 0.829 | 0.662 | 10.79 | 20.26 | 4.88 | 7.23 |
| ICP-Flow | 2 | 3.2m | 0.569 | 0.430 | 0.569 | 0.749 | 0.530 | 8.81 | 17.53 | 3.51 | 5.38 |
| FastNSF | 2 | 2.6m | 0.560 | 0.436 | 0.523 | 0.737 | 0.543 | 12.16 | 18.20 | 6.11 | 12.18 |
| *Feed-forward* | | | | | | | | | | | |
| SeFlow | 2 | 6s | 0.544 | 0.396 | 0.635 | 0.726 | 0.419 | 8.19 | 16.15 | 3.97 | 4.45 |
| VoteFlow | 2 | 8s | 0.538 | 0.355 | 0.605 | 0.780 | 0.410 | 7.80 | 15.65 | 3.51 | 4.24 |
| SeFlow++ | 3 | 7.5s | 0.509 | 0.327 | 0.583 | 0.716 | 0.409 | 6.13 | 14.59 | 1.96 | 1.86 |
| TeFlow (Ours) | 5 | 7s | **0.395** | **0.303** | **0.461** | **0.474** | **0.344** | **4.64** | **10.92** | 1.49 | 1.51 |

ensure fairness, Argoverse 2 results are obtained directly from the public leaderboard, and nuScenes baselines are reproduced following OpenSceneFlow[1], using best reported training configurations.

## 5.1 State-of-the-art Comparison

TeFlow achieves state-of-the-art accuracy on both Argoverse 2 and nuScenes while maintaining real-time efficiency, as shown in Table 1 and Table 2 respectively. On Argoverse 2 test set, TeFlow achieves a Three-way EPE of 3.57 cm, on par with the best optimization-based method Floxels, while being 150× faster (8 s vs 24 min). On Dynamic Bucket-Normalized EPE, TeFlow improves by 22.3% overall compared to SeFlow++, with consistent gains across all categories, including a 31% error reduction for pedestrians. On nuScenes validation set, TeFlow again outperforms all baselines. It achieves the best dynamic normalized score (0.395) and the lowest Three-way EPE (4.64 cm), representing a 22.4% improvement over SeFlow++. The most significant advance is the 33.8% error reduction for the challenging pedestrian class. Together, these results show that TeFlow delivers optimization-level accuracy while retaining the efficiency and scalability of feed-forward methods, setting a new state-of-the-art for self-supervised scene flow estimation.

## 5.2 Ablation Studies on Design Choices

To further understand the source of performance gains in TeFlow, we conduct ablation studies on Argoverse 2, with results reported in Tables 3 and 4, and more analyses presented in Section B.

**Number of Input Frames** Table 3 ablates the impact of the number of input frames. *Two-frame setting:* To assess the contribution of our formulation, we re-implement SeFlow on the same DeltaFlow

---

[1]https://github.com/KTH-RPL/OpenSceneFlow

Table 3: Ablation on the number of input frames on the Argoverse 2 validation set. All experiments use the same DeltaFlow backbone for a fair comparison. TeFlow surpasses SeFlow even with two frames, and performance peaks at five frames, indicating the optimal temporal window. The best results are shown in **bold**.

| Loss Type | #Frame | Dynamic Bucket-Normalized ↓ | | | | | Three-way EPE (cm) ↓ | | | |
|---|---|---|---|---|---|---|---|---|---|---|
| | | Mean | CAR | OTHERS | PED. | VRU | Mean | FD | FS | BS |
| SeFlow | 2 | 0.408 | 0.319 | 0.412 | 0.369 | 0.531 | 6.35 | 16.63 | 1.48 | 0.92 |
| TeFlow | 2 | 0.353 | 0.271 | 0.389 | 0.329 | 0.424 | 5.98 | 13.93 | 2.53 | 1.46 |
| | 4 | 0.283 | 0.204 | 0.342 | 0.295 | 0.293 | 4.57 | 10.77 | 1.87 | 1.08 |
| | 5 | **0.265** | **0.198** | **0.275** | 0.295 | **0.293** | **4.43** | **10.36** | 1.86 | 1.08 |
| | 6 | 0.269 | 0.197 | 0.305 | 0.290 | 0.284 | 4.55 | 10.66 | 1.87 | 1.12 |
| | 8 | 0.300 | 0.269 | 0.336 | **0.273** | 0.321 | 5.40 | 13.50 | 1.78 | 0.91 |

Table 4: Ablation study of proposed self-supervised loss items. Results are evaluated on the Argoverse 2 validation set with default hyperparameter. **Bold** indicates the best performance and red highlights settings with a significant performance drop.

| Loss item | | | Dynamic Bucket-Normalized ↓ | | | | | Three-way EPE (cm) ↓ | | | |
|---|---|---|---|---|---|---|---|---|---|---|---|
| $\mathcal{L}_{gemo}$ | $\mathcal{L}_{static}$ | $\mathcal{L}_{dcls}$ | Mean | CAR | OTHER | PED. | VRU | Mean | FD | FS | BS |
| ✓ | | | 0.386 | 0.317 | 0.586 | 0.297 | 0.343 | 8.85 | 17.26 | 4.45 | 4.85 |
| ✓ | ✓ | | 0.458 | 0.321 | 0.654 | 0.481 | 0.377 | 6.37 | 17.15 | **1.25** | **0.73** |
| | | ✓ | 0.303 | 0.254 | 0.310 | **0.285** | 0.362 | 8.53 | 12.28 | 7.17 | 6.14 |
| | ✓ | ✓ | 0.313 | 0.233 | 0.402 | 0.296 | 0.321 | 4.84 | 11.99 | 1.73 | 0.80 |
| ✓ | ✓ | ✓ | **0.265** | **0.198** | **0.275** | 0.295 | **0.293** | **4.43** | **10.36** | 1.86 | 1.08 |

backbone with an identical two-frame input. TeFlow achieves a 13.5% reduction in dynamic EPE (0.353 vs. 0.408), mainly due to our candidate pool and the cluster-level dynamic loss term, which provides more consensus information and ensures balanced supervision across object sizes. *Multi-frame setting:* Expanding the temporal window within TeFlow to five frames yields the best performance, lowering dynamic EPE by 24.9% to 0.265. This performance gain can be explained by Figure 1b: the multi-frame supervision produced by TeFlow closely follows the ground truth and is more stable than the fluctuating signals from two-frame supervision. Training with these stable signals results in significantly better performance. Further extending the number of frames shows little help or even degrades the performance, which is consistent with the prior findings in the supervised method Kim et al. (2025); Zhang et al. (2025b). A possible explanation is that overly distant frames introduce noisy or less relevant motion, outweighing the benefits of a longer context.

**Self-supervised Loss Item** Table 4 evaluates the contribution of each loss term in our proposed self-supervised objective. Using only the geometric loss provides limited supervision, as nearest-neighbor alignment provides coarse motion cues. Adding the static term improves three-way EPE but increases the dynamic normalized error. Training with only the proposed dynamic-cluster loss $\mathcal{L}_{dcls}$ achieves strong dynamic performance, especially for pedestrians, since the temporal ensembling discovers reliable supervision from multi-frame consistency; however, the absence of static constraints leads to large errors in static regions (FS, BS). Combining $\mathcal{L}_{static}$ with $\mathcal{L}_{dcls}$ restores balanced accuracy, while incorporating all three losses delivers the best result (0.265), reducing the dynamic normalized error by 31.3% compared to the geometric baseline and demonstrating that our multi-frame self-supervised objective effectively unifies geometric, static, and dynamic cues into a consistent training signal.

## 5.3 ANALYSIS ON HYPERPARAMETER SELECTION

We further analyze the sensitivity of TeFlow to its key hyperparameters by varying one parameter at a time while keeping the others fixed at their optimal values. Results are reported in Table 5 and provide additional insight into the functioning of the temporal ensembling strategy.

**Top-K** This parameter controls the number of external candidates in the candidate pool. A small, high-quality set proves most effective, with the best performance at $K = 5$. Larger values introduce noise from less reliable geometric matches and degrade accuracy.

**Cosine Similarity** This threshold determines which candidates are included in the consensus matrix. The optimal value of 0.707 (45°) strikes the right balance: looser thresholds allow inconsistent motions, while stricter ones discard valid candidates too early.

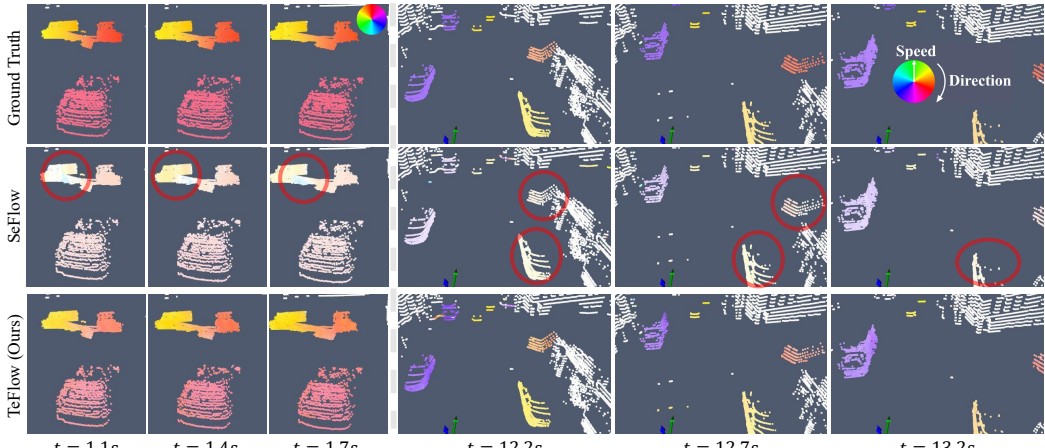

Figure 3: Qualitative results on Argoverse 2 (left) and nuScenes (right). Rows show ground truth, SeFlow, and TeFlow predictions across time. Scene flow is visualized with hue indicating direction and saturation representing speed. Compared to SeFlow, TeFlow produces flow estimates that are more accurate and temporally consistent, particularly for dynamic objects (red circles).

Table 5: Ablation study on the key hyperparameters of TeFlow, evaluated on the Argoverse 2 validation set. The default and best-performing configuration is cosine similarity $\tau_{cos} = 0.7$ (45°), Top-K = 5, and time decay $\gamma = 0.9$. In each row, only the specified parameter is varied from this setting.

| TeFlow Setting | Dynamic Bucket-Normalized ↓ | | | | Three-way EPE (cm) ↓ | | | |
|---|---|---|---|---|---|---|---|---|
| | Mean | CAR | OTHER | PED. | VRU | Mean | FD | FS | BS |
| Default | **0.265** | **0.198** | **0.275** | 0.295 | **0.293** | **4.43** | **10.36** | 1.86 | 1.08 |
| $\tau_{cos} = 0$ (90°) | 0.307 | 0.239 | 0.365 | 0.291 | 0.332 | 5.19 | 13.02 | 1.60 | 0.95 |
| $\tau_{cos} = 0.9$ (20°) | 0.289 | 0.207 | 0.356 | 0.294 | 0.297 | 4.42 | 10.41 | 1.80 | 1.04 |
| $K = 20$ | 0.353 | 0.283 | 0.355 | 0.312 | 0.463 | 5.88 | 14.97 | 1.68 | 1.00 |
| $K = 10$ | 0.307 | 0.241 | 0.314 | 0.296 | 0.377 | 5.11 | 12.39 | 1.83 | 1.12 |
| $\gamma = 1$ | 0.303 | 0.224 | 0.348 | 0.311 | 0.330 | 4.73 | 11.55 | 1.66 | 0.98 |
| $\gamma = 0.5$ | 0.285 | 0.232 | 0.308 | **0.290** | 0.311 | 4.92 | 11.65 | 1.98 | 1.12 |

**Time Decay** This factor weights candidates by their temporal distance, giving higher importance to recent frames. Our default of $\gamma = 0.9$ outperforms both no decay ($\gamma = 1.0$) and stronger decay ($\gamma = 0.5$). Without decay, distant frames are treated equally and introduce noise, while overly strong decay underutilizes longer-term consistency that benefits large, predictably moving objects.

## 5.4 QUALITATIVE RESULTS

Figure 3 presents qualitative comparisons on two challenging dynamic scenarios. On the left (Argoverse 2), the scene contains a moving car and an articulated truck making a turn. The cab and trailer of the truck exhibit distinct motions that are visible in the ground truth, but SeFlow fails to capture them and predicts a uniform flow across the entire vehicle. TeFlow, in contrast, models the articulated components more accurately, producing flows that closely match the ground truth. On the right (nuScenes), a multi-object scene is shown. Here, estimates for the moving vehicle (red circles) in SeFlow are unstable and flicker across frames, while TeFlow delivers stable and temporally consistent flow throughout the sequence. More qualitative examples are provided in Section C.

## 6 CONCLUSION

In this work, we introduced TeFlow, a self-supervised feed-forward approach that unlocks the benefits of multi-frame supervision for real-time scene flow estimation. By mining temporally consistent supervisory signals through our temporal ensembling and voting strategy, TeFlow overcomes the limitations of traditional two-frame supervision and unstable point-wise correspondences. On the Argoverse 2 and nuScenes benchmarks, TeFlow sets a new state-of-the-art for self-supervised, real-time methods, improving accuracy by up to 33%. It successfully closes the gap with slower optimization-based approaches, offering comparable performance at a 150x speedup, thereby achieving both high accuracy and efficiency.

**Ethics Statement**   The research presented in this paper adheres to the ICLR Code of Ethics. Our work focuses on scene flow estimation, a fundamental task in 3D perception for autonomous systems. The datasets used for training and evaluation, Argoverse 2 and nuScenes, are large-scale public datasets that have been properly anonymized and are widely used by the research community. Our method aims to improve the accuracy and robustness of perception systems, which could contribute to enhancing the safety of autonomous vehicles and other robotic applications. We do not foresee any direct negative societal impacts or ethical concerns arising from this work.

**Reproducibility Statement**   We are committed to ensuring the reproducibility of our research. To this end, we will release the complete source code, training configurations, and all pre-trained model weights used to generate the results in this paper. A detailed description of the implementation, including key hyperparameters and training infrastructure, is provided in Section 4.3. The datasets used are publicly available, and we describe our data processing steps to allow for a faithful reproduction of our experimental setup. We believe these resources will enable the community to easily verify our results and build upon our work.

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

# A   DATASETS DESCRIPTION

The Argoverse 2 dataset is a primary benchmark for scene flow estimation, consisting of 700 training scenes, 150 validation scenes, and 150 test scenes, totaling approximately 107,000 annotated training frames. Our main evaluations are conducted on the official test split, with results compared against the Argoverse 2 Scene Flow Challenge leaderboard Khatri et al. (2024), which provides official baseline results. For the local validation, we follow Zhang et al. (2025b) and apply dynamic motion compensation to generate ground-truth flow labels.

The nuScenes dataset contains 700 training and 150 validation scenes and is also used in our evaluation. Since nuScenes does not provide official scene flow annotations, we follow the protocol of Zhang et al. (2025a) to generate pseudo ground truth. To ensure a consistent temporal resolution, the native 20Hz LiDAR data is first downsampled to 10Hz, resulting standard 100ms interval between frames. For each object, a rigid transformation is estimated from its 3D bounding box annotations and instance ID. This transformation is then applied to all LiDAR points within the object to compute their displacements, which serve as pseudo ground-truth flow labels. These labels are generated only for the validation set, while training uses the full 137,575 unlabeled frames, demonstrating the scalability of our self-supervised approach.

For both datasets, ground points are removed prior to evaluation. In Argoverse 2, we use the provided HD maps following the official protocol, whereas in nuScenes we apply a line-fitting-based ground segmentation method Himmelsbach et al. (2010). As a result, all reported evaluation metrics are computed exclusively on non-ground points.

# B   ADDITIONAL QUANTITATIVE ANALYSIS

**Importance of Internal Candidate** Table 6 examines the role of the internal candidate in our temporal ensembling strategy (Section 4.1). When only external candidates are used, supervision relies solely on geometric correspondences between frames, which are often unstable under occlusion or sparse observations, leading to higher errors (0.321 vs. 0.265). By including the internal candidate predicted by the network, the supervisory pool gains a stable reference that anchors learning and suppresses noisy geometric matches. This combination reduces the dynamic normalized error by 17.4% and improves three-way EPE across all categories, confirming that the internal candidate is essential for producing reliable consensus supervision and stable training.

Table 6: Ablation on the importance of internal candidates. Results are evaluated on the Argoverse 2 validation set with five input frames. Including internal candidates provides a stable reference for the voting scheme and consistently improves the performance.

| Candidates Pool | Dynamic Bucket-Normalized ↓ | | | | Three-way EPE (cm) ↓ | | | |
|---|---|---|---|---|---|---|---|---|
| | Mean | CAR | OTHER | PED. | VRU | Mean | FD | FS | BS |
| Only External | 0.321 | 0.278 | 0.403 | 0.281 | 0.321 | 5.42 | 13.53 | 1.74 | 0.98 |
| Both (Proposed) | **0.265** | **0.198** | **0.275** | **0.295** | **0.293** | **4.43** | **10.36** | 1.86 | 1.08 |

Table 7: Ablation study of dynamic cluster loss $\mathcal{L}_{\text{dcls}}$. Results are evaluated on the Argoverse 2 validation with 5 input frames. The other two loss item kept unchanged. **Bold** indicates the best performance and red highlights settings with a significant performance drop.

| $\mathcal{L}_{\text{dcls}}$ formulation | Dynamic Bucket-Normalized ↓ | | | | Three-way EPE (cm) ↓ | | | |
|---|---|---|---|---|---|---|---|---|
| | Mean | CAR | OTHER. | PED. | VRU | Mean | FD | FS | BS |
| Only Point-level | 0.351 | 0.258 | 0.331 | 0.352 | 0.463 | 4.92 | 12.66 | **1.29** | **0.80** |
| Only Cluster-level | 0.356 | 0.222 | 0.603 | 0.284 | 0.316 | 5.31 | 12.99 | 1.87 | 1.09 |
| Both (Proposed) | **0.265** | **0.198** | **0.275** | **0.295** | **0.293** | **4.43** | **10.36** | 1.86 | 1.08 |

**Dynamic Cluster Loss Formulation** Table 7 evaluates the contribution of the point-level and cluster-level terms in the proposed dynamic cluster loss $\mathcal{L}_{\text{dcls}}$ (Equation (9)). Training with only the point-level term underperforms on small and slow-moving agents such as pedestrians and VRUs,

supporting our claim that point-wise supervision is dominated by large clusters containing many points. When trained with only the cluster-level term, the model improves small-object performance but loses fine-grained alignment for large dynamic objects, resulting in an 82% increase in the OTHER category error (0.603 vs. 0.331). Combining both terms achieves the best overall performance, reducing the dynamic normalized error by 24.5% and 25.6% compared to the point-level and cluster-level variants, respectively, demonstrating the effectiveness of our proposed self-supervised formulation in providing reliable and balanced supervision across different object scales.

**Different Multi-frame Backbone** Table 8 evaluates the generality of our self-supervised framework on different multi-frame backbones. When adopting Flow4D, the model already benefits from multi-frame temporal reasoning but achieves a mean dynamic normalized error of 0.330. Replacing it with the $\Delta$Flow backbone yields consistent improvements across all categories, reducing the overall dynamic error by 19.7% (0.330 to 0.265) and three-way EPE by 22.3% (5.70 to 4.43). This trend aligns with results observed in supervised training, where $\Delta$Flow provides more effective temporal representation and motion modeling. These results verify that our proposed self-supervised objective is agnostic to the backbone architecture and can be seamlessly applied to future multi-frame scene flow networks as they emerge.

Table 8: Ablation study on different multi-frame backbones within our self-supervised pipeline. Results are evaluated on the Argoverse 2 validation set with five input frames. The results show that our self-supervised objective consistently improves performance across two distinct multi-frame backbones (Flow4D and $\Delta$Flow), indicating that the method is architecture-agnostic and readily applicable to future multi-frame scene flow networks.

| Backbone | Dynamic Bucket-Normalized ↓ | | | | | Three-way EPE (cm) ↓ | | | |
|---|---|---|---|---|---|---|---|---|---|
| | Mean | CAR | OTHER | PED. | VRU | Mean | FD | FS | BS |
| Flow4D | 0.330 | 0.254 | 0.326 | 0.329 | 0.411 | 5.70 | 12.98 | 2.67 | 1.46 |
| $\Delta$Flow | **0.265** | **0.198** | **0.275** | **0.295** | **0.293** | **4.43** | **10.36** | 1.86 | 1.08 |

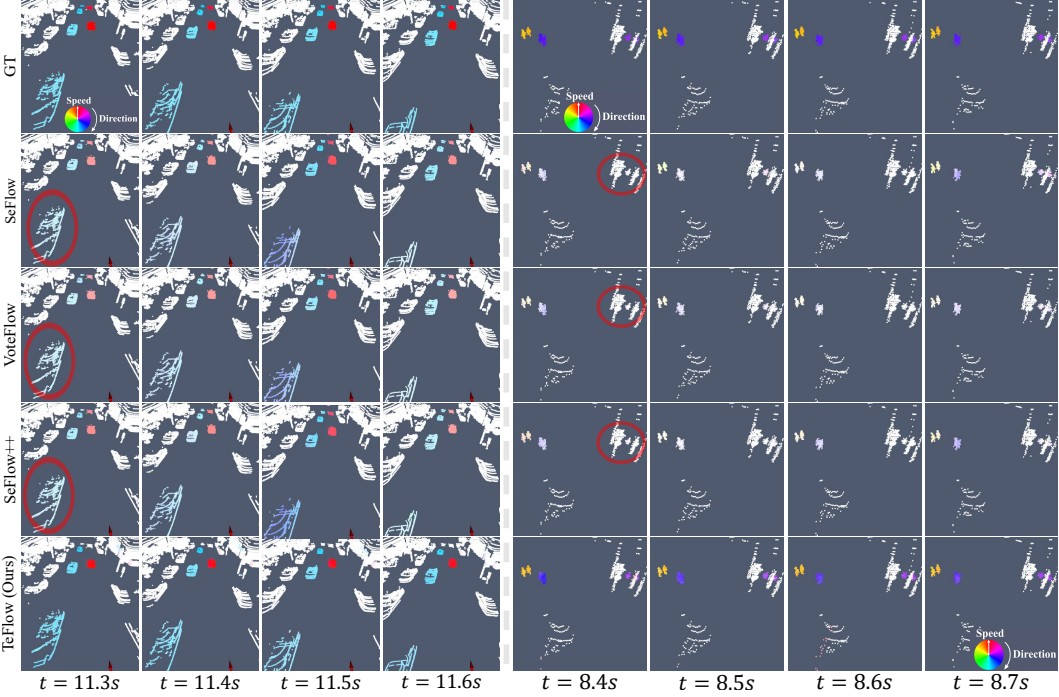

| | | | | | | | | |
|---|---|---|---|---|---|---|---|
| $t = 11.3s$ | $t = 11.4s$ | $t = 11.5s$ | $t = 11.6s$ | $t = 8.4s$ | $t = 8.5s$ | $t = 8.6s$ | $t = 8.7s$ |

Figure 4: Qualitative comparisons on the Argoverse 2 validation set. Left: A multi-vehicle scene. Right: A vehicle stopping for pedestrians. Our method robustly handles both scenarios, unlike the baseline. (Best viewed in color.) The scenes correspond to scene IDs 'c85a88a8-c916-30a7-923c-0c66bd3ebbd3' and 'b6500255-eba3-3f77-acfd-626c07aa8621'.

## C  QUALITATIVE RESULTS

The qualitative results in the main paper are derived from the scenes '8749f79f-a30b-3c3f-8a44-dbfa682bbef1' and 'scene-0104' in the Argoverse 2 and nuScenes validation set, respectively.

Here, we present additional qualitative results comparing our TeFlow with top self-supervised feed-forward methods, namely SeFlow Zhang et al. (2024b), VoteFlow Lin et al. (2025), and Se-Flow++ Zhang et al. (2025a). All visualizations use a standard color-coding scheme, where hue indicates motion direction and saturation encodes speed.

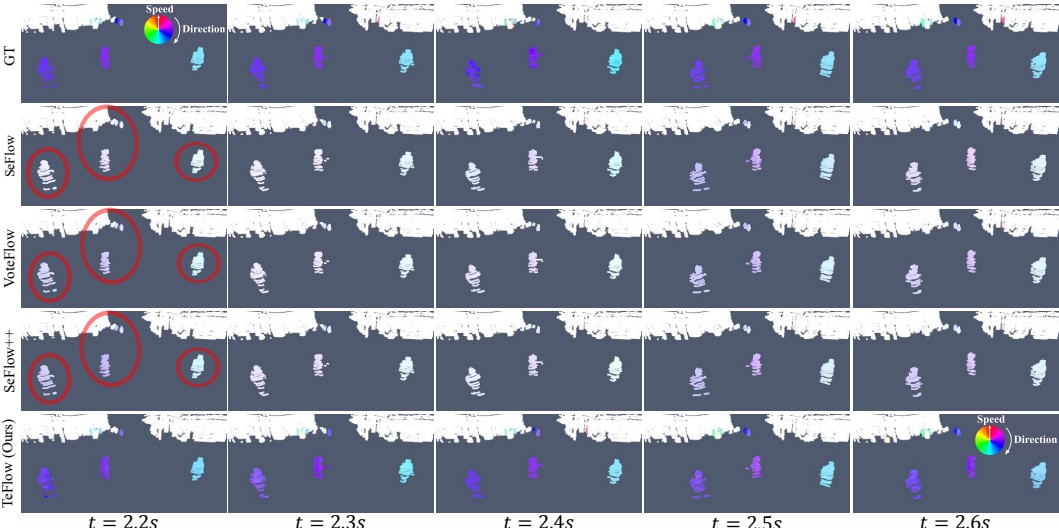

Figure 5: Qualitative results on the Argoverse 2 validation set. Our method accurately captures the motion of multiple pedestrians, while all feed-forward baselines underestimate the flows of moving pedestrians. (Best viewed in color.) The scenes correspond to scene IDs '9f871fb4-3b8e-34b3-9161-ed961e71a6da'.

Figure 4 shows two complex multi-agent scenes from Argoverse 2. In the left scene, three oncoming vehicles are captured. While the ground truth indicates consistent forward motion, all baseline feed-forward methods occasionally predict conflicting directions (e.g., flows shift from blue to purple around $t = 11.3$–$11.5$), reflecting the instability of two-frame supervision. In contrast, our TeFlow maintains coherent motion across time, producing stable and accurate flow for each vehicle. The right scene highlights another common failure case: pedestrians motion. The ground truth reveals clear trajectories, including a distant pedestrian partially occluded by a lamp post. Baseline methods consistently underestimate the flow magnitudes of these small or occluded agents, resulting in weak or inconsistent predictions. While our TeFlow captures their motion with the correct magnitude and direction.

Figure 5 presents a challenging scene with three pedestrians crossing the road simultaneously. In the ground truth, all pedestrians exhibit clear motion, yet baseline feed-forward methods underestimate their flow magnitudes due to noisy two-frame supervision, resulting in weak and inconsistent predictions under such dynamic motion. In contrast, the model trained with our TeFlow objective produces flow fields that are both spatially coherent and temporally stable. Each motion of pedestrian is captured with accurate magnitude and direction, closely matching the ground truth across the time window. Furthermore, TeFlow also preserves reliable estimates for other small or distant dynamic objects, highlighting its robustness under challenging scenarios with sparse observations.

Figure 6 shows a challenging scene from the nuScenes validation set. In the lower-left corner, five pedestrians are walking together, while a vehicle and another pedestrian are passing in front of the ego car. The ground truth indicates clear motion for both the vehicle and pedestrians. However, baseline feed-forward methods significantly underestimate the vehicle's flow magnitude and often fail to detect the motions of the smaller pedestrians. In contrast, TeFlow produces a smooth and com-

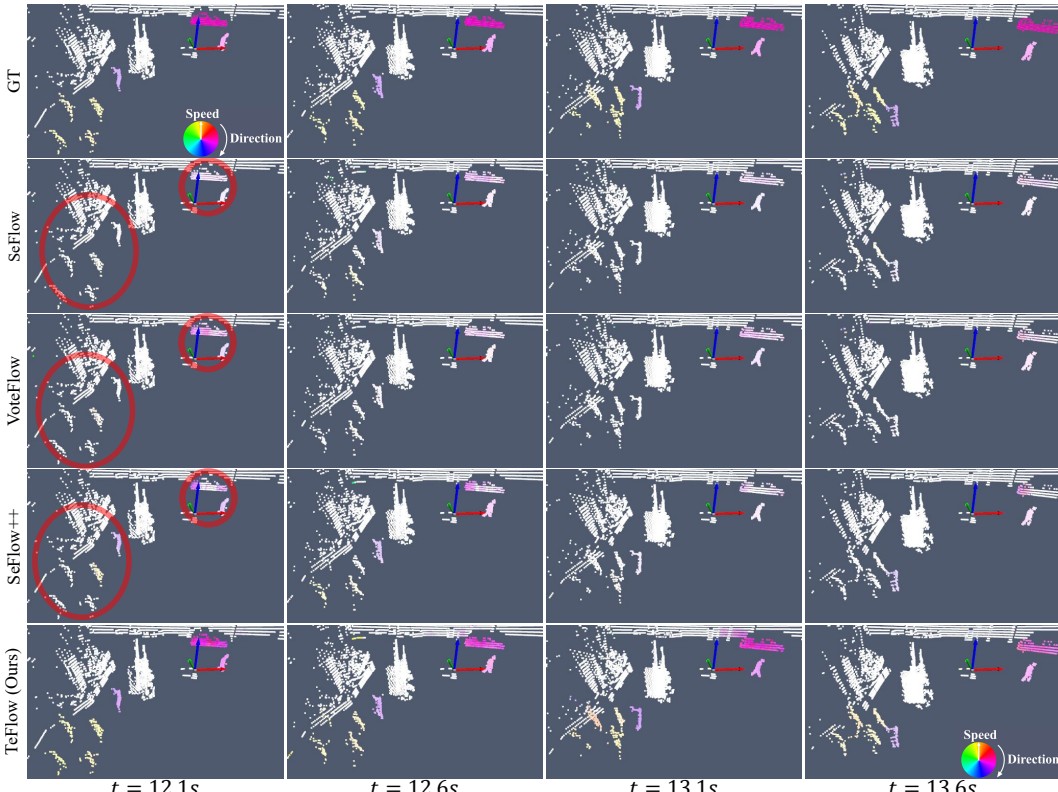

Figure 6: Qualitative results on the nuScenes validation set. On this sparser data, TeFlow provides complete motion for the vehicle and detects the pedestrians, whereas the baseline underestimates the car's flow and misses the smaller actors. (Best viewed in color.) The scenes correspond to the scene IDs 'scene-0025'.

plete flow field for the vehicle and successfully captures the individual motions of the pedestrians, even under the sparse point density of nuScenes.

Figure 7 illustrates a complex roundabout scene from the Argoverse 2 validation set. Multiple vehicles are moving along curved trajectories. The baseline methods fail to provide consistent estimates, often underestimating the motion or producing fragmented flows, especially for vehicles entering or exiting the roundabout. While, TeFlow produces coherent and smooth flow fields that closely follow the ground-truth directions, demonstrating its ability to handle complex multi-agent interactions in curved motion scenarios.

## D    OTHER DISCUSSION

**Detail Description of LLM Usage** We utilized a Large Language Model (LLM) only as a writing assistant for language polishing and grammar checking. The authors retained full control of the manuscript, and all scientific content, ideas, methods, and experiments were entirely conceived, executed, and written by the authors.

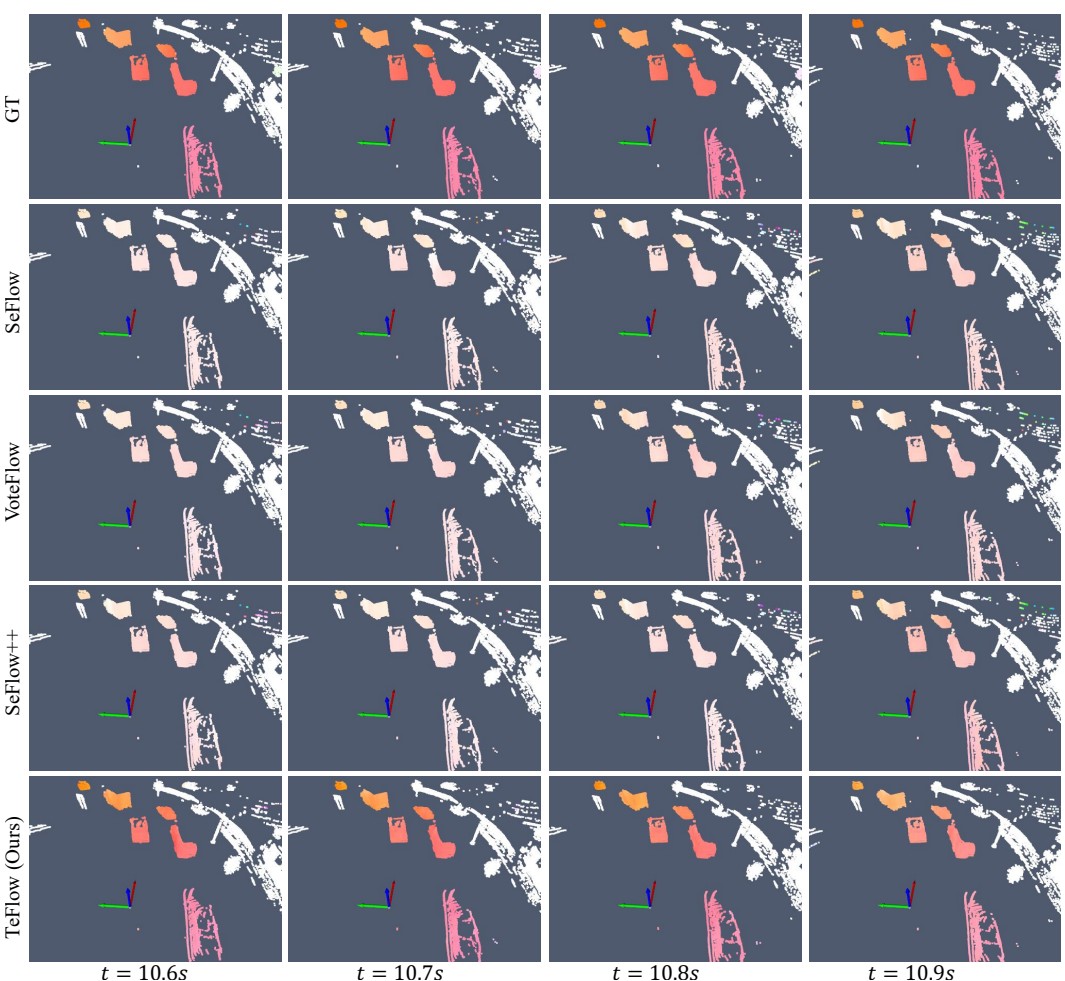

Figure 7: Qualitative results on the Argoverse 2 validation set. Our method accurately captures the motion of vehicles in complex roundabout scenarios. (Best viewed in color.) The scenes correspond to scene IDs 'bdb9d309-f14b-3ff6-ad1f-5d3f3f95a13e'.

