# OpenReview forum: "TeFlow:  Enabling Multi-frame Supervision for Feed-forward Scene Flow Estimation"
_ICLR.cc/2026/Conference — ICLR 2026 Conference Withdrawn Submission_

### Official Review · Reviewer_AJPB · 2025-10-30

**Soundness:** 2
**Presentation:** 2
**Contribution:** 2
**Rating:** 4
**Confidence:** 4

**Summary:**

This paper investigates self-supervised scene flow estimation from multi-frame point clouds. It introduces a self-supervised framework that segments the scene into static and dynamic regions, and leverages temporal ensembling and voting to obtain supervision signals for the dynamic parts. Experimental results on the Argoverse 2 and nuScenes datasets show that the proposed approach achieves competitive performance with low computational cost.

**Strengths:**

- The proposed approach demonstrates competitive performance compared to other feed-forward methods on the Argoverse 2 and nuScenes datasets.
- The experimental evaluation is comprehensive.

**Weaknesses:**

1. The writing should be improved.

- Figure 2 needs improvement.
  As the key figure illustrating the overall framework, Figure 2 does not effectively help readers understand the temporal ensembling and voting algorithms. In particular, the meanings of the different colors and arrows in the motion candidate pool are not explained, making it difficult to interpret the figure.

- The writing of Section 4.1 should be improved.
  In Line 213, the paper states that "we establish correspondences by finding, for each point p_i, its nearest neighbor in P," whereas in Eq. (3), the nearest-neighbor search is performed between p_k and P. This makes the process of motion candidate generation hard to follow.

2. The rationale of Motion Candidate Generation needs to be further clarified.

    When generating the supervisory signal from previous frames, the method directly finds the nearest neighbor of the current points in the previous frame, without warping the current points according to the (predicted) motion between the two frames. By ignoring the inter-frame motion, performing nearest-neighbor search without such warping or motion compensation becomes inappropriate for establishing accurate correspondences. It is worth noting that in self-supervised scene flow estimation, almost all self-supervised loss functions (e.g., Chamfer loss) warp the source points toward the target frame to find correspondences and thereby generate the supervision signal.

3. In Eq. (5), the authors use the motion direction (i.e., cosine similarity) to measure the consistency between two flow candidates. It would be helpful to explain why the end point error (EPE) is not used. Since cosine similarity only accounts for the direction of motion while ignoring its magnitude, the consistency evaluation may be incomplete or potentially misleading.

**Questions:**

1. Please explain the detailed process of Motion Candidate Generation, especially Line 213 and Eq. (3).
2. Please clarify the rationale behind the design of Motion Candidate Generation.
3. Why is cosine similarity used instead of the EPE for measuring the consistency? Is there any experimental evidence supporting this design choice?

---

> ### Author Response · Authors · 2025-11-12
>
> Thanks for the valuable comments! We have addressed your concerns and questions individually below.
>
> ---
>
> [**W1**: Writing Confusion.]
>
> (Figure 2) In the Motion Candidate Pool (left), green and gray points denote dynamic points observed at different time frames, while each arrow represents one motion hypothesis obtained by connecting a point in the current frame to its Top-K farthest nearest neighbors in other frames (Eq. 4). These arrows together form a pool of motion candidates $\mathcal{F}\_j$ for each dynamic cluster.
> The red arrows indicate the consensus winner and its agreeing neighbors—those motion candidates that pass the directional consistency check and contribute to the final supervision according to Eqs. (7)–(8).
> On the right, the matrix M encodes pairwise directional consistency (1 = consistent, 0 = inconsistent), and w denotes the reliability weights combining temporal decay and motion magnitude. The aggregated output $\bar{\mathbf{f}}_{\mathcal C\_j}$ represents the reliable cluster-level flow used for supervision. To improve clarity, we will revise Figure 2 and its caption.
>
> (Section 4.1) Line 213 and Eq. (3) refer to two sequential steps in motion candidate generation. Line 213 describes the initial nearest-neighbor search, where each point ($p\_i$) in the cluster finds its nearest neighbor in another frame. This produces a set of raw correspondences. Eq. (3) then applies *after* this step: we select the Top-K correspondences with the largest displacements, and the source points of these selected pairs are denoted as ($p\_k$) in Eq. (3). Thus, Eq. (3) defines the motion candidates computed from the Top-K subset rather than contradicting Line 213. We will clarify this connection in the revised manuscript.
>
> ---
>
> [**W2**: Rationale behind the design of Motion Candidate Generation.]
>
> The nearest-neighbor search in our motion candidate generation is not intended to produce direct correspondences for supervision, but rather to construct a diverse candidate pool of motion hypotheses. These hypotheses capture possible geometric displacements across frames without assuming a specific motion model or relying solely on the current network prediction.
> In contrast to Chamfer-based losses, which require warping the source points using the predicted flow, our goal is to obtain model-agnostic geometric evidence that can later be refined through temporal ensembling and reliability voting. This aggregation step automatically suppresses inconsistent or noisy candidates and emphasizes temporally consistent motions.
> Empirically, combining our designed motion candidates with reliability-based voting and aggregation reduces the impact of prediction errors from warping and leads to more stable supervision signals, as evidenced by the consistent improvements over Chamfer-only baselines (Table 4).
>
> ---
>
> [**W3**: Why is cosine similarity used instead of the EPE for measuring the consistency?]
>
> We intentionally use cosine similarity to assess directional consistency, rather than EPE to assess magnitude similarity, for two reasons:
>
> 1) *Magnitude consistency* is not the objective: In our design, larger displacements are treated as stronger signals (cf. Top-K largest distances in candidate generation), and their influence is modeled explicitly by the reliability weights in Eq. (6). If we also required candidates to be consistent in length, a binary consensus mask would systematically favor small/weak motions (which happen to be numerically closer), suppressing the strong cues we want to emphasize. Moreover, a 0/1 threshold on distances is brittle for continuous, scale-dependent magnitudes (sensitive to scene density, occlusion, and per-frame variability), and does not reflect whether two motions agree semantically.
>
> 2) Direction captures the agreement we need; magnitude is handled elsewhere: Direction provides the stable component of motion across frames, while magnitude is already accounted for in the reliability term (Eq. (6)) that up-weights clear/large motions. This separation (direction for consensus in Eq. (5), magnitude for reliability in Eq. (6)) prevents weak motions from dominating the consensus and yields more stable aggregation.

---

### Official Review · Reviewer_qK36 · 2025-10-30

**Soundness:** 3
**Presentation:** 3
**Contribution:** 2
**Rating:** 4
**Confidence:** 5

**Summary:**

The paper presents a feed-forward network that learns how to solve scene flow using temporal ensembling strategy. The results are strong and on part of the dataset examines showed significant improvement over STOA. The used technique involves adding temporal data and a joined cost function over points and blocks.

**Strengths:**

The primary strength of the TeFlow method  is its introduction of cluster loss that enables balanced multi-frame supervision. While prior feed-forward methods rely on two-frame correspondence losses, TeFlow first aggregates a highly stable and temporally consistent motion target for each dynamic object cluster through a temporal ensembling strategy. This cluster-level averaging prevents the loss from being dominated by larger objects with more points, ensuring that smaller dynamic objects, such as pedestrians, receive fair and effective supervision.

**Weaknesses:**

The ideas presented in this paper are not new but their combination provides strong outcome. Specifically, clustering of object-level loss enforcement was already published (and cited by the authors), as well as temporal constraints (more than two frames). Hence, while they provided solution is worthy and achieve STOA in some cases, it is an incremental improvement over known methods.

**Questions:**

Please elaborate on the contribution of each item already used and known in literature over the provided solution.

---

> ### Author Response · Authors · 2025-11-12
>
> Thanks for your comments. We have addressed your concerns and questions individually below.
>
> ---
>
> [**W1**: Incremental Improvement.]
>
> We respectfully disagree with the assessment that TeFlow is an incremental combination of existing ideas.
> Although clustering-based losses and temporal cues have indeed appeared in previous literature, these techniques were designed for either multi-frame supervised pipelines or two-frame self-supervised training.
>
> However, *none of these ideas can be directly used to train a multi-frame self-supervised feed-forward model*, because frame-to-frame geometric signals are highly inconsistent under occlusion and sparsity (Fig. 1b). Applying these prior temporal or cluster-based objectives naively leads to unstable or unusable supervision. This is precisely the gap TeFlow addresses.
>
> **Our contribution is not combining existing components, but making multi-frame self-supervision *possible* for feed-forward models.**
>
> TeFlow introduces a **temporal ensembling mechanism** that, to our knowledge, is the *first* solution enabling stable multi-frame training in a feed-forward setting. Our pipeline:
>
> 1. builds a motion candidate pool across multiple frames,
> 2. evaluates the candidates via directional consensus and magnitude-driven reliability, and
> 3. aggregates a stable cluster-level target flow through weighted voting.
>
> This mechanism resolves the key barrier that prevented previous self-supervised feed-forward methods (including SeFlow and VoteFlow) from benefiting from multi-frame inputs. As also shown in our experiments, directly adopting prior multi-frame Chamfer losses or cluster-based losses in feed-forward training yields unstable supervision and fails to improve accuracy (see Table 4 and Appendix B).
>
> **Impact of the proposed mechanism**
>
> The resulting multi-frame supervision is significantly more stable than all two-frame baselines (Fig. 1b), enabling substantial performance improvements (See Tables 1-3):
>
> * **33% error reduction** compared to SOTA feed-forward methods.
> * consistent gains across all dynamic classes on Argoverse 2 and nuScenes.
> * accuracy on par with leading optimization-based pipelines, **150× faster**.
>
> Therefore, TeFlow is not a simple combination of known components but a new enabling technique that makes multi-frame self-supervised training feasible for feed-forward scene flow estimation, a capability that has not been available in the literature.

---

> ### Author Response · Authors · 2025-11-14
>
> [**Q1**: Please elaborate on the contribution of each item already used and known in literature over the provided solution.]
>
> Below, we summarize how each item, while related to prior work, contributes concretely within our proposed solution, and we integrate this clarification into the revised paper.
>
> - Temporal Ensembling and Candidate Voting: The concept of leveraging temporal constraints is known, but existing applications are confined to *either two-frame self-supervised methods or fully multi-frame supervised pipelines*. TeFlow introduces a new, specifically designed temporal ensembling mechanism, the *first* solution enabling stable, effective multi-frame supervision in a feed-forward setting. This mechanism aggregates motion candidates across multiple frames and uses agreement and reliability to select a consensus flow. It resolves the instability issue (highlighted in Fig. 1b). Ablations on the number of frames (Tab. 3), internal vs. external candidates (Tab. 6) and backbone variants (Tab. 8) show that our temporal-ensembling mechanism consistently improves performance and works reliably across the tested multi-frame backbones.
>
> - Dynamic-cluster loss ($\mathcal{L}\_{\text{dcls}}$): Prior self-supervised methods (e.g., SeFlow) used only point-level consistency terms to supervise motion and operated solely on two frames.  In contrast, we formulate a new loss, $\mathcal{L}_{\text{dcls}}$, extends this by introducing a cluster-level term alongside the point-level term (Eq. 9), in order to prevent large objects from dominating the optimization and ensure fair supervision across objects of different sizes.
> Furthermore, $\mathcal{L}\_{\text{dcls}}$ operates on multi-frame supervisory signals, built upon the reliable output of our core temporal ensembling mechanism. This stable input enables the loss to jointly enforce coherent motion at both fine (point) and object (cluster) scales. The combined loss is essential for the large improvements seen on dynamic categories, where previous geometric or static terms alone could not produce comparable results (Tab. 7).
>
> - Geometric Loss ($\mathcal{L}\_{\text{geom}}$): This term follows the standard self-supervised geometric alignment used in prior optimization-based work, serving to provide a necessary coarse motion signal. In our ablations (Tab. 4), $\mathcal{L}\_{\text{geom}}$ serves as the essential, yet weak, baseline upon which the efficacy of all our novel components is measured.
>
> - Static Regularization ($\mathcal{L}\_{\text{static}}$): This term implements the established static-consistency constraints found in the literature. While it improves FS/BS EPE (Tab. 4), its main contribution is to complement the dynamic-cluster loss, ensuring balanced accuracy and recovering stability once the multi-frame dynamics are robustly modeled by our new mechanism.
>
> In summary, the most significant contribution of this paper is the design of a Dynamic Multi-Frame Self-Supervised Mechanism and its associated loss ($\mathcal{L}\_{\text{dcls}}$). Although this work uses the same high-level concepts as all scene-flow estimation (temporal constraints, object-scale focus), *TeFlow is the first framework that enables these benefits within a multi-frame self-supervised feed-forward learning paradigm*. We conclusively demonstrate that our dynamic loss is the driving force behind the substantial performance improvements, as previous loss formulations are inefficient in this new multi-frame setting.

---

### Official Review · Reviewer_s9pS · 2025-10-31

**Soundness:** 3
**Presentation:** 3
**Contribution:** 3
**Rating:** 6
**Confidence:** 3

**Summary:**

This work introduces a supervised multi-frame scene flow prediction framework. To address issues such as occlusion and multi-frame temporal expansion, the proposed TeFlow presents an effective temporal aggregation strategy, according to the authors, which has significant speed improvements and performance advantages.

**Strengths:**

1. Good presentation and clear writng, which makes it easy to read.

2. Effective method design and good performance.

3. Comprehensive Experimental Validation. The study includes rigorous evaluations on two large-scale autonomous driving datasets, with detailed ablation studies on input frame count, loss components, and hyperparameters.

**Weaknesses:**

1. Although the method is leading in many metrics, it can be learned that TeFlow has room for improvement on some indicators, which are areas that can be done better.

2. Line 464, inconsistent capitalization.

3. Has the speed of this method been averaged from multiple measurements? Specifically, how many times?

**Questions:**

Please refer to the weaknesses.

---

> ### Author Response · Authors · 2025-11-12
>
> Thanks for the valuable comments! We have addressed your concerns and questions individually below.
>
> ---
>
> [**W1**: Although the method is leading in many metrics, it can be learned that TeFlow has room for improvement on some indicators, which are areas that can be done better.]
>
> In the comparison with other feed-forward methods (Table 1), TeFlow achieves the best performance across all dynamic categories and in the primary leaderboard metric, the three-way EPE. This metric is widely used because it balances foreground dynamic (FD), foreground static (FS), and background static (BS) performance. By contrast, methods that classify most points as static (for example, ego-motion flow) can achieve perfect FS and BS scores while performing extremely poorly on dynamic regions.
>
> Regarding FS and BS specifically, TeFlow indeed has slightly higher errors in these static categories, but the differences are very small (approximately 3 mm). In comparison, TeFlow reduces the dynamic EPE (FD) by 240–370 mm over competing feed-forward methods, and improves the three-way EPE by 80–340 mm. These gains dominate the minor differences in static regions and reflect the fact that dynamic motion estimation is substantially more challenging. Dynamic motion estimation is also more important for autonomous driving.
>
> ---
>
> [**W2**: Inconsistent capitalization]
>
> Thanks for pointing this out. We've revised in the updated version.
>
> ---
>
> [**W3**: Speed calculation]
>
> Yes, the reported runtime is an averaged measurement. For each dataset, we compute the per-scene runtime over the full evaluation split. This corresponds to roughly 150 scenes for Argoverse 2 (about 157 frames per scene) and 150 scenes for nuScenes (about 200 frames per scene). We report the mean over all scenes.

---

### Note · Authors · 2025-11-14

I have read and agree with the venue's withdrawal policy on behalf of myself and my co-authors.